# Metaverse and Healthcare: Machine Learning-Enabled Digital Twins of Cancer

**DOI:** 10.3390/bioengineering10040455

**Published:** 2023-04-07

**Authors:** Omid Moztarzadeh, Mohammad (Behdad) Jamshidi, Saleh Sargolzaei, Alireza Jamshidi, Nasimeh Baghalipour, Mona Malekzadeh Moghani, Lukas Hauer

**Affiliations:** 1Department of Stomatology, University Hospital Pilsen, Faculty of Medicine in Pilsen, Charles University, 32300 Pilsen, Czech Republic; 2Department of Anatomy, Faculty of Medicine in Pilsen, Charles University, 32300 Pilsen, Czech Republic; 3Faculty of Electrical Engineering, University of West Bohemia, 30100 Pilsen, Czech Republic; 4Department of Computer Engineering, Mashhad Branch, Islamic Azad University, Mashhad 9187147578, Iran; 5Dentistry School, Babol University of Medical Sciences, Babol 4717647745, Iran; 6Department of Radiation Oncology, Medical School, Shahid Beheshti, University of Medical Sciences, Teheran 1985717443, Iran

**Keywords:** breast cancer, digital twins, cancer, machine learning, artificial intelligence, metaverse, healthcare

## Abstract

Medical digital twins, which represent medical assets, play a crucial role in connecting the physical world to the metaverse, enabling patients to access virtual medical services and experience immersive interactions with the real world. One serious disease that can be diagnosed and treated using this technology is cancer. However, the digitalization of such diseases for use in the metaverse is a highly complex process. To address this, this study aims to use machine learning (ML) techniques to create real-time and reliable digital twins of cancer for diagnostic and therapeutic purposes. The study focuses on four classical ML techniques that are simple and fast for medical specialists without extensive Artificial Intelligence (AI) knowledge, and meet the requirements of the Internet of Medical Things (IoMT) in terms of latency and cost. The case study focuses on breast cancer (BC), the second most prevalent form of cancer worldwide. The study also presents a comprehensive conceptual framework to illustrate the process of creating digital twins of cancer, and demonstrates the feasibility and reliability of these digital twins in monitoring, diagnosing, and predicting medical parameters.

## 1. Introduction

In addition to all common methods in diagnosing and treating cancer, new methods are adapted to increase the cumulative progress of treatment plans and procedures [1,2]. Cancer digital twins are a recent method that processes input data by a variety of AI techniques and biological methods and accordingly represents precise therapy protocols [3,4]. ML methods are considered significant analytical techniques since authentic and reliable results are obtained, whereas profound learning about input data is not required [5,6,7]. While Deep Learning (DL) methods have recently performed major roles during pandemic management, the demand for these methods has been proved in the medical system. Among different analytical systems, intelligence techniques are considered simple whilst they perform powerful data extraction and analyses by classification, pattern recognition, and estimation from huge and complex sources of data. For this reason, ML is increasingly utilized in different fields, including economics, healthcare, engineering, etc. [8]. Smart manufacturing and modernizing industries provided a new concept called “digital twins,” that concentrates on representing “physical twins” by the creation of digital forms of objects, processes, devices, equipment, and components. Physical twins are reconstructed with all errors, processes, and interactions in the real world by dynamization of physical parts in digital form. For this reason, the digital twin is considered fundamental in attracting attention to cyber-physical systems (CPSs) [9]. Nowadays, our society, industries, and any aspect of our lives have been digitalized from the real world to the virtual world by metaverse conceptual technology, which has high potential for the evolution of different services [8]. The new technology of the metaverse allows for the synthesis of virtual and actual assets through cyberspace, enabling people to represent themselves through their avatars [10]. Above all, this technology is highly potent while aided by other techniques and technologies, such as artificial intelligence, transfer learning, deep learning, digital twins, the IoT, edge computing, and cloud computing. Nevertheless, the metaverse requires more enterprises, consideration, and research in the medical field, particularly in the diagnosis, treatment, and examination of cancer. It has significantly developed and progressed in other platforms, including social media, diagnosis, and effective treatment planning, while it has reduced in medication costs. The metaverse combines the virtual and physical worlds, allowing people and their digital representations to interact in a technology-driven environment that includes augmented and virtual reality, mixed and artificial intelligence, digital twins, blockchain, extended reality, and high-speed internet. This technology has recently been used in entertainment and social media, but it has the potential to revolutionize healthcare and significantly impact overall health and clinical practice [7]. The metaverse is a virtual world that is created by combining various advanced technologies and virtual spaces. It has the potential to significantly change the way we work and live and could revolutionize industries, including healthcare. The immersive and interactive features of the metaverse allow healthcare professionals to connect with patients in a more personalized and engaging way, leading to better care and patient satisfaction. The metaverse can also help facilitate the sharing of information and resources, enabling more efficient and effective treatment in the healthcare industry. Overall, the metaverse has the potential to significantly improve and transform the healthcare industry [11]. The technology for creating the metaverse is constantly improving and allows users to fully experience the high levels of interaction and immersion offered by the metaverse. Companies are beginning to explore the potential of the metaverse and how it can be incorporated into their existing business models. For example, Italy’s top soccer division recently announced that the AC Milan vs. Fiorentina game will be screened within the Nemesis metaverse, allowing fans to interact within the virtual room for the Serie A league [12]. Traditionally, healthcare has relied on face-to-face interactions between patients and medical professionals for diagnosis, treatment, and surgery. While telehealth has slightly modified this approach, recent technological developments have the potential to significantly transform the healthcare industry. The metaverse could revolutionize healthcare in various ways, including virtual health, mental health, reality management, and virtual management [13].

The potential use of digital twins in healthcare is a new and growing research area that could lead to more individualized patient care [14,15,16]. Personalized treatment planning, including the use of precision medicine, is a well-known aspect of cancer care. The introduction of digital twins in healthcare could potentially support and enhance this personalized approach. The authors have studied the medical literature to understand the best way to introduce digital twins in healthcare. The review specifically focuses on the current use of digital twins in the healthcare field [14].

In this paper, we introduce a technique for constructing digital representations of cancer and their use in the medical sector. The second section provides an extensive explanation of what cancer digital twins are and how they can be utilized to simulate the diagnosis and development of cancer over time. The third section details the proposed cancer digital twin, including the machine learning techniques and algorithms applied to its creation. The fourth section showcases the outcomes of our experiments, exhibiting the ability of the proposed cancer digital twin to accurately predict the behavior of cancer. In the fifth section, we discuss the limitations and potential difficulties associated with using cancer digital twins, as well as the future opportunities of this technology. Finally, the sixth section concludes our work and emphasizes its significance in the area of cancer diagnosis and treatment.

## 2. Cancer Digital Twins

The process of diagnosing and treating patients, particularly in the context of cancer treatment, involves multiple steps and can have certain limitations. These points will be discussed further in the article. Digital twins have the potential to address some of the limitations of the current treatment planning process and improve care in the field of cancer and precision medicine as a whole [17].

The usual method for predicting lung cancer involves using diagnostic criteria such as patient symptoms, physical indications, and unidentified causes [18]. Some studies have suggested that using a combination of patient symptoms, electrocardiograms, conventional d-dimer tests, spiral CT scans of the lungs, and pulmonary angiography can help to predict lung cancer with pulmonary embolism [19].

Laboratory models created from clinical samples are a useful tool in preclinical research for testing drugs and studying disease mechanisms. By combining these laboratory models with their digital counterparts, or predictive mathematical models, digital twins can be created. These digital twins are essential for fully utilizing their potential to address the complexity of the human body at the molecular level. Cell metabolism is closely connected to all other cellular processes, so any disruption in cellular physiology often affects metabolic profiling. In our laboratory, as well as others, changes in metabolism have been shown to play a causal role in the development of cancer. However, the altered metabolic profile of all cancer cells cannot be described by a single metabolic program due to various factors, such as genetic variation and the different nutrient consumption and metabolism needs of various subclones within a tumor [20].

Wickramasinghe et al. proposed a classification system for digital twins based on systems and mathematical modeling theory, dividing them into Grey Box, Surrogate, and Black Box models. They then discussed the use of Black Box digital twins in personalized uterine cancer care as one possible approach. This article presents one of the first uses of digital twins in this context, bringing together three key areas: clinical care, digital health, and computer science [5].

### 2.1. Digital Twins

A digital twin is a virtual replica of a physical object that is connected to the original through a system of operations. The virtual twin should closely mimic the behavior of the physical object, providing real-time data and allowing for the timely identification and resolution of errors. Ultimately, the digital twin should be able to sustain itself and improve the physical model through the incorporation of digital intelligence and iterative improvement. Although there has been a limited number of papers published on digital twins in the past decade compared to their importance, their versatility makes them a cutting-edge technology with the potential to have a significant impact in many areas [21,22].

Digital twins are virtual replicas of physical entities, such as people, devices, objects, systems, and even places that are created using advanced technologies, like analytics, AI, and IoT. They are developed through the combination and optimization of these technologies. Digital twins allow professionals to simulate actions on physical entities before they are carried out in reality. There are numerous business applications for digital twins in three sectors: healthcare, automotive, and manufacturing. The rapid proliferation of IoT sensors is a major factor in the growth of digital twins. These virtual replicas allow for real-time monitoring of physical entities, reducing maintenance burdens by providing early warning of issues. As an example, Chevron has been able to save millions of dollars in maintenance costs through the use of digital twins in its oil fields and refineries. Siemens uses digital twins to model and prototype products that have not yet been manufactured, leading to fewer defects and a faster time to market [19].

Although digital twins have gained popularity more quickly in some industries, such as manufacturing, they have not been as widely adopted in the service sector, including healthcare. However, the use of digital twins could bring significant benefits to clinical care, particularly in cancer treatment, thanks to advances in precision medicine and increased computational capabilities and analytics [17].

### 2.2. Healthcare Digital Twin

A healthcare digital twin is a virtual representation of a person that utilizes a lifetime of data and AI-powered models to predict the individual’s health status and provide recommendations for clinical questions. These digital replicas can be used to predict the outcome of certain procedures, such as lung cancer diagnosis [19]. The deployment of intelligent Medical Internet of Things (MIoT) and healthcare digital twins has been steadily increasing and is expected to continue to grow, with an estimated market value of USD 75.44 billion in the next decade [7,23].

### 2.3. Machine Learning Digital Twins

Several studies have used machine learning to predict lung cancer. In one study, an artificial neural network was used to assess the risk of deep vein thrombosis and pulmonary embolism based on 31 risk factors in 294 inpatients. The network was optimized to predict the patient’s condition and reduce the need for perfusion scanning. Another study developed a decision analysis model for CT imaging of adult pulmonary embolism patients using a network to predict whether the patients needed to undergo CT examination. The model transformed original electronic medical record data into a timeline feature vector to improve the usefulness of CT scans for pulmonary embolism patients [24].

## 3. The Proposed Cancer Digital Twin

Despite the many digital twin plans currently in progress across various industries, including healthcare systems, there is a lack of information on how to successfully implement digital twins for complex and nonlinear systems, specifically in the field of medical data which is highly sensitive, complex, and uncertain. Additionally, it is crucial for the system to be easy to use and adaptable for those without extensive knowledge in computer science or biomedical engineering, such as nurses and doctors, as they will be the ones using, developing, and modifying the system. The proposed digital twin platform uses ML to collect and analyze healthcare data for each patient, creating a multimodal dataset for various ML methods. This dataset includes information on behavioral, microbial, metabolic, immune, epigenetic, and genetic issues, covering all aspects of the problem. The platform, as shown in Figure 1, also includes health reports with clinical, biological, and psychological information. These medical data are then transferred to the metaverse to update the digital twins of cancer patients, aiding in finding the best treatment method or helping physicians make informed decisions. In the metaverse, a variety of multimodal ML models are utilized to process the data from the real world. It is important to note that each individual patient should have their specific digital twin for their specific problem, allowing for continuous monitoring by the most advanced expert systems in the metaverse. With major companies putting many resources into developing the metaverse platform, healthcare applications are set to improve significantly. This can result in a big change in the way cancers are diagnosed and treated through digitalizing patient data.

### 3.1. Case Study—Breast Cancer

As a case study, the proposed approach has been applied to breast cancer (BC), which is the second most common form of cancer and affects a large number of people worldwide. BC is classified as a malignant tumor that develops from breast cells and can affect both women and men [25]. Although the dataset used in this study is specific to female breast cancer, the approach can also be applied to male breast cancer. Breast cancer is a complex disease that can affect many organs in the body, including the breasts, lymph nodes, and even distant organs such as the lungs or bones. Early diagnosis is crucial for a successful course of treatment and can greatly improve the chances of survival for patients. The proposed approach aims to provide additional screening tools and robust prediction models based on data from routine consultations and blood analysis. These models can assist in determining if the breast cancer index has changed, which can help physicians make more informed decisions about treatment options. The use of digital twinning and machine learning in this case study demonstrates the potential of this technology in revolutionizing the diagnosis and treatment of breast cancer, providing more accurate and efficient care for patients.

### 3.2. Participants

The dataset used in this study is related to cases of breast cancer (BC) and was collected from patients who were recruited from the University Hospital Centre of Coimbra (CHUC) between 2009 and 2013. The diagnosis of BC was confirmed through cancer histology and positive mammography for each individual patient. The data were collected before any treatment or surgery, and individuals who had already started treatment were removed from the dataset to ensure that the data represents a pre-treatment status. In addition, a group of healthy women also volunteered as controls and were included in the study. These controls were chosen based on the criteria of not having a prior cancer treatment and being healthy in terms of other diseases and infections. It is important to have a control group to compare the data of healthy individuals with the data of individuals who have BC. All participants provided consent for the study, which was approved by the Ethical Committee of CHUC. The research focused on assessing metabolic dysregulation and hyperresistinemia, which are metabolic disorders that are commonly found in cancer patients. The dataset included 52 healthy participants and 64 women with BC. This dataset provided a significant sample size to conduct a thorough analysis and draw meaningful conclusions.

### 3.3. Sample Analysis

The dataset used in this study was collected by taking blood samples and demographic, anthropometric, and clinical data from patients who were diagnosed with breast cancer (BC) [25]. The samples were taken after overnight fasting and were collected on the same day to ensure that the conditions were similar for all patients. The dataset includes information on menopausal status, height, weight, and age of the patients.

Body mass index (BMI) was calculated by dividing the weight by the squared height of the patients. This information helps to understand the overall health status of the patient. In addition to the above information, various parameters were measured at the University of Coimbra. With the use of some assay kits, Chemokine Monocyte Chemoattractant Protein 1 (MCP-1), Resistin, Adiponectin, and Leptin were evaluated. These parameters are important as they are related to the metabolic status of the patients, which can affect the course of the treatment and the prognosis of the disease. The Homeostasis Model Assessment (HOMA) and plasma levels of insulin were also embedded in this dataset. These parameters help to understand the glucose metabolism and insulin sensitivity of the patients. All of this information is crucial in the diagnosis and treatment of breast cancer.

## 4. Results

The digital twin of breast cancer (BC) was created using the case study outlined above, to be the main component of the proposed platform. Although the dataset used is limited and is not enough to create robust and precise ML-based digital twins, the goal of this study is to demonstrate how ML techniques can be used to create these models. The digital twins created with this dataset are not comprehensive enough to be used in medical applications directly, but they can serve as a starting point for further research and development. In order to use these digital twins in medical applications, more reliable and comprehensive clinical data must be added through the IoT. This will give us more accurate and detailed information about the patients, which can be used to improve the diagnosis and treatment of BC. For this process, three parameters—MCP-1, Resistin, and Adiponectin—were considered as targets, and the remaining parameters were used as inputs for the ML models in the training process. The digital twin is not only a replica of the patient but also a tool for medical staff to simulate treatment options and evaluate the outcomes. In this way, the digital twin can assist medical staff to make better decisions and improve the care for patients with BC.

### 4.1. ML Linear Regression

The first ML method that was used to create the digital twin of cancer symptoms based on the dataset is ML Linear Regression [26,27]. Figure 2 demonstrates the diagnosis of cancer using a linear statistical method that models the relationship between a scalar response (dependent variable) and at least one explanatory variable (independent variable) the figure shows two bands, green and red, for each patient and method. The green band represents patients who are healthy in terms of breast cancer, while the red band represents patients with this cancer. The magnitude of the figure increases with the severity of cancer. It is important to note that this is not a binary classification system but a multiclass classification of patients. While our proposed digital twin model shows promise in accurately diagnosing cancer, it is important to note that advanced monitoring systems are necessary for commercial use. Nonetheless, our focus in this research paper is on the design and methodology of the digital twin model, rather than its user-friendliness. As with any groundbreaking research, further development and refinement are required to optimize its commercial potential. The orange graph in Figure 3 depicts the actual values of the aforementioned parameters while the dashed blue graph illustrates the predicted values of MCP-1, Leptin, Resistin, and Adiponectin.

Based on Figure 2, In the case that there is just one explanatory variable, the method is known as simple linear regression; if there are more, it is known as multiple linear regression. This method is different from multivariate linear regression, which is used for predictions of multiple correlated dependent variables, as opposed to a single scalar variable. Linear regression uses linear predictor functions to model relations with the model’s unknown parameters being estimated from the data. These models can be utilized for digital twinning because the digital version of the systems needs to recognize complexity or accuracy. Most frequently, it is assumed that the conditional mean of the response when considering the values of the explanatory variables (or predictors) is an affine function of those values; the conditional median or another quantile is applied less frequently.

Similar to all other methods of regression analysis, the concentration of linear regression is on the conditional probability distribution of the response based on the values of the predictors rather than the joint probability distribution of all these variables, which is the focus of multivariate analysis. In this case, the MCP-1, Leptin, Resistin, and Adiponectin parameters are the dependent variable and the model is trying to fit a line that best represents the relationship between the MCP-1, Leptin, Resistin, Adiponectin parameters and the independent variables. According to Figure 3, the closer the orange graph is to the dashed blue graph, the better the model performed in estimating the MCP-1, Leptin, Resistin, and Adiponectin parameters. Additionally, this figure demonstrates the performance of the used method by testing data. Therefore, these figures are more significant than the training performance.

### 4.2. Decision Tree Regression

The second machine learning method that has been employed to create a digital twin of a patient’s status based on an existing dataset is Decision Tree Regression (DTR) [28,29]. This supervised learning approach, called a decision tree, is widely used in a range of regression modeling and is particularly useful for analyzing complex datasets. A decision tree is a type of graph that uses true or false responses to specific queries to categorize or regress data. When visualized, it resembles a tree with different types of nodes at the root, internal, and leaf layers. The decision tree begins at the root node and branches out to internal nodes and leaf nodes. The final classification categories or actual values are found in the leaf nodes. The structure of the tree is determined by the features of the dataset, and the process of creating a decision tree is called decision tree induction. Choosing a feature to serve as the root node is the first step in creating a decision tree. Figure 4 and Figure 5 illustrate the model performance. Figure 6 and Figure 7 show graphical representations of the decision-making process of a prediction model.

As Figure 4 shows, the diagnosis of cancer utilizing a linear statistical technique that models the association between a scalar response (dependent variable) and one or more explanatory variables (independent variables). The figure displays two bands, green and red, for each patient and method. The green band corresponds to patients who are free of breast cancer, whereas the red band corresponds to those with this condition. The severity of cancer increases with the magnitude of the figure. It is worth noting that the system is not based on binary classification but rather on multiclass classification of patients. Although our digital twin model displays potential in accurately diagnosing cancer, sophisticated monitoring systems are required for commercial use. However, in this research paper, our emphasis is on the design and methodology of the digital twin model, rather than its user-friendliness. As with any pioneering research, further development and improvement are necessary to optimize its commercial viability.

Additionally, Figure 5 displays the performance of the method being used with testing data, which is more important than the performance during training. In order to determine the Gini impurity for a feature with numerical values, the data must first be sorted in ascending order, and then the averages of the adjacent values are computed. The Gini impurity is then determined at each chosen average value by organizing the data points according to whether the feature values are smaller or larger than the chosen value and whether the selection accurately categorizes the data. In addition to being a powerful method for regression modeling, decision trees are also easy to interpret and understand, which makes them a popular choice for digital twinning. The ability to modify the decision tree based on the dynamic of the data further increases its usefulness in this application. Figure 6 and Figure 7 illustrate the implementation of the method.

### 4.3. Random Forest Regression

We employed Random Forest Regression (RFR) as an additional machine learning method in the creation of our data-driven digital twin. RFR is a widely used technique for both classification and regression problems in machine learning [30,31,32]. One of the main benefits of using this method is its simplicity and accuracy. These attributes make it a suitable option for physicians to use in building a digital twin of cancer. In Figure 8, a RFR technique is used to diagnose cancer by analyzing the relationship between a scalar response (dependent variable) and one or more explanatory variables (independent variables). Figure 9 illustrates the performance for RFR.

As it can be seen in Figure 8, the figure has two bands, green and red, for each patient and method. The green band shows patients who do not have breast cancer, while the red band shows those who do. The figure’s magnitude increases with the severity of cancer. It is important to note that the system does not rely on a binary classification, but instead on a multiclass classification of patients.

Although our digital twin model has the potential to accurately diagnose cancer, advanced monitoring systems are necessary for commercial use. However, in this research paper, we focus on the design and methodology of the digital twin model, rather than its user-friendliness. As with any groundbreaking research, further development and improvement are needed to make it commercially viable.

According to Figure 9, the method’s performance. As can be seen from the figure, the results are very promising and demonstrate the potential of using RFR in digital twin creation. The simplicity of RFR also makes it easy to interpret and understand the results, which is a significant advantage in the medical field where accuracy and transparency are of the utmost importance. Overall, RFR is a powerful and effective method that can be utilized in the creation of data-driven digital twins of cancer, and we believe it has the potential to revolutionize the way we approach cancer treatment. Creating the digital twin of the proposed disease, we used Random Forest Regression (RFR) as our third machine learning method. RFR is a commonly used algorithm for regression problems because of its ease of use and high level of accuracy.

RFR is an ensemble technique that combines multiple decision trees and uses a voting mechanism to make predictions. It is more versatile than Decision Tree Regression (DTR) because of the randomness added by the algorithm. This helps to reduce the variance of the model. RFR is typically trained using the bagging technique, which combines predictions from various machine learning algorithms to provide predictions that are more accurate than those from a single model. It does not require extensive parameter adjusting and is less susceptible to dataset outliers. The only RFR parameter normally required for experimentation is the number of ensemble trees. The predictions are calculated as the average prediction across all decision trees. The fact that the separate models have little association with one another is crucial. RFR uses a voting mechanism to produce predictions based on decision trees. It creates multiple decision trees by dividing the training samples. In accordance with the bootstrap sampling method, a portion of the data set is randomly chosen as the training example, and the remaining data are utilized as the validation sample for each decision tree. To obtain the final predictions while regressing unknown samples, the predictions of each decision tree are first generated, and all of the predictions are then aggregated using a simple voting procedure.

### 4.4. Gradient Boosting Algorithm

As the last proposed machine learning method, we employed the Gradient Boosting Algorithm (GBA) for digitalizing cancer based on the dataset [33,34]. The GBA is a powerful supervised machine learning technique that has been proven to be effective in analyzing and addressing complex problems. This is why it is considered a robust and reliable backbone for medical analysis. The GBA is an iterative algorithm that improves the model by focusing on the difficult cases which other algorithms might have struggled with. This allows the GBA to deliver accurate and high-performing models, which is particularly useful in the medical field, where accuracy is of the utmost importance.

The proficiency of the technique in identifying breast cancer is presented in Figure 10, using both the training and testing datasets. The results of the digitalization of cancer performed by this machine learning technique are shown in Figure 11.

As can be seen from the figure, the results are very promising and demonstrate the potential of using the GBA in digitalizing cancer. The GBA is a highly sophisticated algorithm that can handle large datasets and can be adapted to various medical applications, making it a valuable tool for medical analysis and research. Overall, the GBA is a powerful and effective method that can be utilized in the digitalization of cancer and we believe it has the potential to revolutionize the way we approach cancer diagnosis and treatment.

As technology continues to advance, the various subfields of computational methods in modeling and controlling systems have been multiplying and adapting. One of the key areas that have seen significant growth is the use of machine learning methods, which help to compute the signals generated by these methods with the highest rate of accuracy. This is evident in fields such as healthcare, where machine learning is being used to analyze and interpret large amounts of patient data in order to improve diagnosis and treatment.

In this research, a new approach to digital twinning cancer using machine learning has been presented. The important contribution of this research is to indicate how machine learning methods can be used to construct standard data-driven digital twins that can be used in the metaverse for making decisions about patient treatment or ensuring precise diagnosis. This way of modeling cancer has several advantages over traditional methods. First and foremost, since the measured healthcare data contain all nonlinearities and uncertainties, making decisions about sensitive issues is more reliable and accurate when using machine learning. Additionally, cancers exhibit a wide range of complex behaviors over a long period of time, which can be difficult to accurately model using mathematical models alone. By using machine learning, we are able to analyze large amounts of data and identify patterns that may not be immediately apparent, which can lead to more effective treatment and better outcomes for patients. Furthermore, digital twinning with machine learning allows us to simulate the progression of cancer over time, and make predictions about how it will behave in the future. This can be extremely valuable in developing new treatments and therapies, as well as in identifying potential complications before they occur.

### 4.5. Discussion

The COVID-19 pandemic has led to an increase in the use of healthcare technologies, including remote patient care and virtual follow-up, which are likely to continue to be adopted as a way to prepare for future pandemics due to the ongoing development of artificial intelligence. In a paper on the subject, the potential uses of digital twin technology in healthcare were discussed. Digital twins involve a physical model, a virtual counterpart, and the interaction between the two, and the intersection of computer science and medicine through the use of digital twins is a new and promising area with many potential applications [21].

Digital twins have greatly impacted manufacturing by enabling new and innovative processes and more personalized and individualized solutions. It is possible that similar benefits could be seen in healthcare if digital twins were adopted and, in some cases, modified or extended for various healthcare applications. Figure 12 illustrates the performance of our prediction model, as evaluated by four common metrics: mean squared error (MSE) [35], root mean squared error (RMSE) [36], mean absolute error (MAE) [37], and coefficient of determination (R2). MSE is a commonly used measure of the difference between the predicted values and the actual values. It calculates the average of the squared differences between the predictions and the true values. The MSE is a measure of the quality of an estimator and it is always non-negative, and values close to zero are better. In this study, we aim to diagnose breast cancer through the creation of a digital twin representation of the disease. This digital twin is based on certain parameters that are crucial for breast cancer diagnosis. We then use binary classification to categorize the digital twin as either having breast cancer or not having breast cancer.

To predict these parameters, we use four different machine learning methods. Each of these methods is designed to determine the value of one of the parameters that make up the digital twin. By combining the results of these four methods, we are able to diagnose breast cancer through the analysis of the digital twin. In simpler terms, we have developed a system that uses advanced algorithms to help diagnose breast cancer. This system takes into account multiple factors that are important for the diagnosis, and combines them to make an accurate determination of whether a patient has breast cancer or not.

Root mean squared error (RMSE) is the square root of the MSE. It is used to measure the difference between the predicted and actual values and is often used for continuous data. It gives an idea of the magnitude of the error and is expressed in the same units as the response variable.

Mean absolute error (MAE) is a measure of the difference between the predicted and actual values. It calculates the average of the absolute differences between the predictions and the true values. Like MSE, MAE is also a measure of the quality of an estimator, it is always non-negative, and values close to zero are better. Coefficient of determination (R2) is a statistical measure that represents the proportion of the variation in the dependent variable that is explained by the independent variable. It ranges between 0 and 1, with values closer to 1 indicating a better fit. A model with a higher R2 value has a better ability to predict the target variable. We discuss how digital twins could be used to support improved personalized and individualized cancer care planning and complement advances in precision medicine by incorporating findings from healthcare data. We propose a novel machine learning-based approach for conceptualizing digital twins in cancer care and provide a use case example of uterine endometrial cancer to demonstrate how this might be applied in practice. We argue that machine learning digital twins are well-suited for personalized and individualized cancer care planning and that the use of digital twins in clinical decision-making, combined with data mining and advances in fields like genome analysis, has the potential to bring about similar revolutionary improvements in healthcare, particularly in cancer care, as have been seen in manufacturing through the use of digital twins.

The creation of digital twins will allow researchers and scientists to use data in a way that meets their specific needs and to employ ICT tools to analyze, study, validate, visualize, and so on. These virtual patients will become a powerful ICT tool that can identify new biomarkers for diseases by aggregating data and allowing doctors and researchers to visualize and interpret the results. Data from patients will be approached from a new perspective using GAN machines, which allow for the complete anonymization of health records and may have an impact on EU data protection laws. This technique will produce flexible and interpretable models that can be used to train new doctors and provide expert doctors with new insights about diseases [38].

A “virtual digital twin” is a digital representation of an individual created using high-resolution data. It is an extension of the “actual digital twin” health model. To provide a truly personalized platform for nutrition, the “virtual digital twin” should include a personal health baseline and allow for the inclusion of health and lifestyle data from different stages of life, such as young and old age. This approach is similar to the proposed longitudinal holo’ome monitoring for colon cancer [39]. An example of how a “virtual digital twin” could be created is through the use of an obesity genetic risk score (GRS) as an early life, albeit incomplete, virtual representation of baseline risk for weight gain. This risk can be augmented over time by incorporating genomic, metagenomic, immune, behavioral, and deep phenotypic data. Integrative -omics technologies, wearable sensors, the Internet of Things (IoT), and consumer internet behavior can provide the necessary data and tools to facilitate early preventive actions for health maintenance [39].

People using the metaverse interact with high-quality, real-time 3D renderings and applications that require a lot of bandwidth and cannot afford delays. The metaverse also relies on data from the physical world, which is collected by IoT devices and sensors. These data are used to update the status of digital objects in the metaverse. To provide users with a smooth and immersive experience, it is necessary to process a large amount of data quickly. Creating and implementing a customized metaverse can be complex and costly, as it requires advanced technologies to process and store the data generated within the metaverse [10].

The digital twin developed in this study can be used in the healthcare system to improve the accuracy and efficiency of breast cancer diagnosis. By creating a digital representation of the disease, the system can provide a more comprehensive understanding of the patient’s condition. By using advanced machine learning methods, the system can quickly and accurately analyze the patient’s data, taking into account multiple factors that are important for diagnosis. This can lead to earlier and more accurate detection of breast cancer, which can have a positive impact on patient outcomes. In practical terms, this means that healthcare professionals can use the digital twin to help diagnose breast cancer faster and more accurately. This can lead to quicker treatment and improved outcomes for patients, and can help reduce the burden on the healthcare system as a whole.

Our research emphasizes the way of creating the digital twins of cancer and its medical ways in a general way, but we also acknowledge the significance of data protection and GDPR compliance in developing medical digital twin platforms. To ensure secure and confidential handling of sensitive patient data, robust data encryption and access control mechanisms can be implemented in the digital twins, allowing only authorized personnel to access the data based on their roles and responsibilities. This way, this platform can benefit from data anonymization techniques to protect patient identities while allowing medical data analysis, especially when integrating data from different hospitals or sources. To facilitate secure and standardized data transmission and processing, standard interfaces and protocols can be implemented. Compliance with GDPR and ISO 27001 standards and regulations related to data privacy and security is also essential to ensure that the digital twin platform adheres to industry standards and best practices. Overall, we primarily focus on the digital twins of cancer, but the importance of GDPR compliance and data protection in medical digital twin platform development can be a good future goal to complete this framework. Robust data encryption, access control mechanisms, data anonymization techniques, standard interfaces and protocols, and adherence to relevant regulations and standards can make the digital twin platform an effective and secure solution for integrating and analyzing medical data from different sources.

## 5. Conclusions

The metaverse, AI, and digital twins have gained popularity in recent years, but their development and implementation in healthcare systems may have some limitations. One significant limitation of the use of the metaverse in healthcare is the transfer of concepts and information from the real world to the virtual environment. Additionally, the complexity of medical disorders makes their digitalization more complicated, potentially impacting the scalability, flexibility, and usability of these emerging technologies in the healthcare system. Recognizing diseases like cancer as serious and requiring constant analysis and monitoring, this research proposes ML-powered methods for digital twinning cancer while taking into account these limitations. These methods include ML Linear Regression (ML LR), Decision Tree regression (DTR), Random Forest Regression (RFR), and Gradient Boosting Algorithm (GBA). By utilizing these machine learning techniques, the platform can analyze and interpret large amounts of patient data and create accurate models of cancer progression while accurately discriminating between healthy and affected individuals. Therefore, using a reputable dataset, several ML-based strategies for breast cancer have been modeled and simulated to demonstrate the feasibility and ease of the digital twinning process. This approach enables us to simulate the diagnosis and progression of cancer over time and make predictions about its future behavior. This can be extremely valuable in developing new treatments and therapies and identifying potential complications before they occur. However, it is important to note that, despite its advantages, the use of ML-based digital twinning in healthcare systems has limitations, such as the requirement for substantial amounts of data for model training, the possibility of bias in both the data and models and the difficulty of interpreting results for non-experts. Nevertheless, by addressing these limitations, the proposed platform has the potential to revolutionize cancer diagnosis and treatment.

## Figures and Tables

**Figure 1 bioengineering-10-00455-f001:**
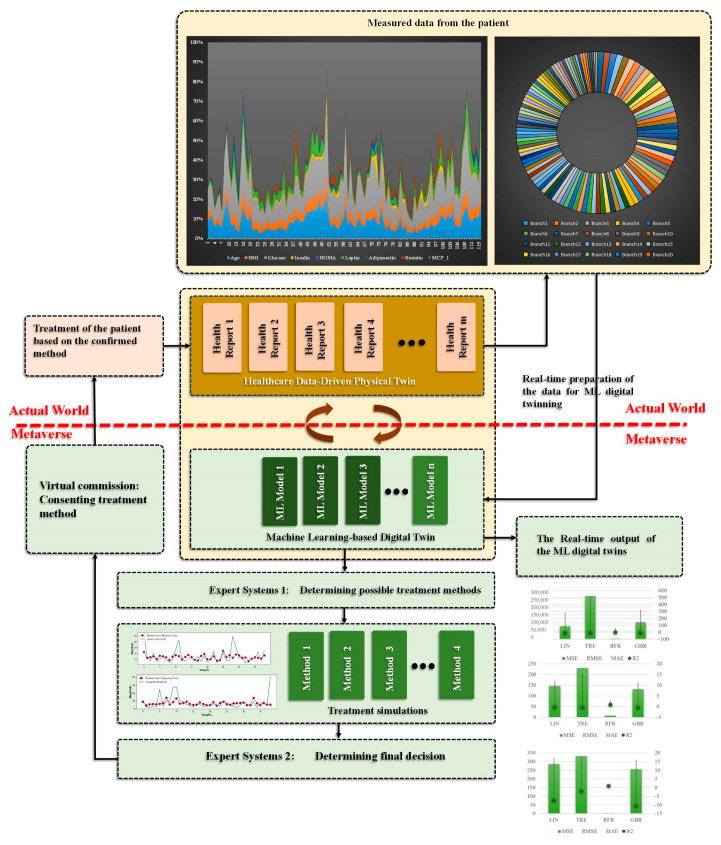
The proposed framework is focused on the concept of digital twinning for cancer patients using machine learning. The framework includes the creation of a physical twin, or a replica of the actual patient who has cancer, that is embedded in the system. This physical twin has real-time interactions with the corresponding digital twin, which is an exact virtual replica of the patient. As the patient undergoes treatment, data are continuously collected and analyzed. The digital twin then uses several ML-based methods in the metaverse, a virtual space, to evaluate the progress of the treatment or the extent of the disease. These ML models are designed to provide fast and robust processing of the measured data, giving physicians a more accurate picture of the patient’s condition. The results obtained from the digital twin are then transferred to a group of physicians in a virtual commission, where they are discussed and evaluated. Based on the interactions and results, the best possible treatment methods are selected. The proposed framework demonstrates the potential of digital twinning and machine learning in revolutionizing the diagnosis and treatment of cancer, providing more accurate and efficient care for patients.

**Figure 2 bioengineering-10-00455-f002:**
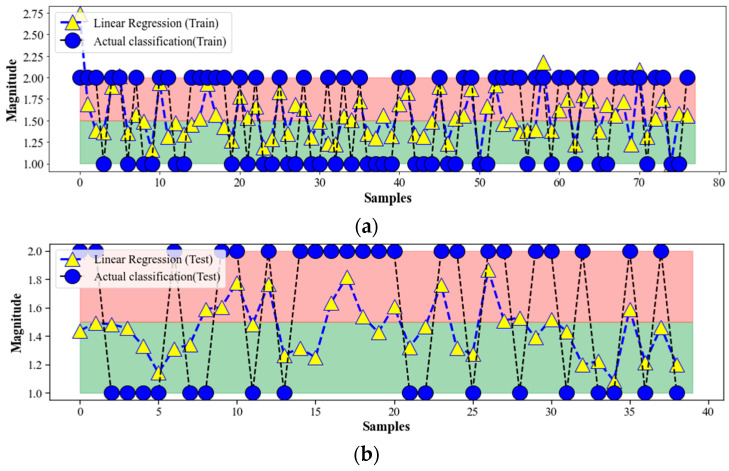
The procedure of identifying breast cancer is illustrated for both the training and testing data based on ML linear regression. (**a**) The effectiveness of the procedure is measured using the training data. (**b**) The procedure’s performance is assessed using the testing data.

**Figure 3 bioengineering-10-00455-f003:**
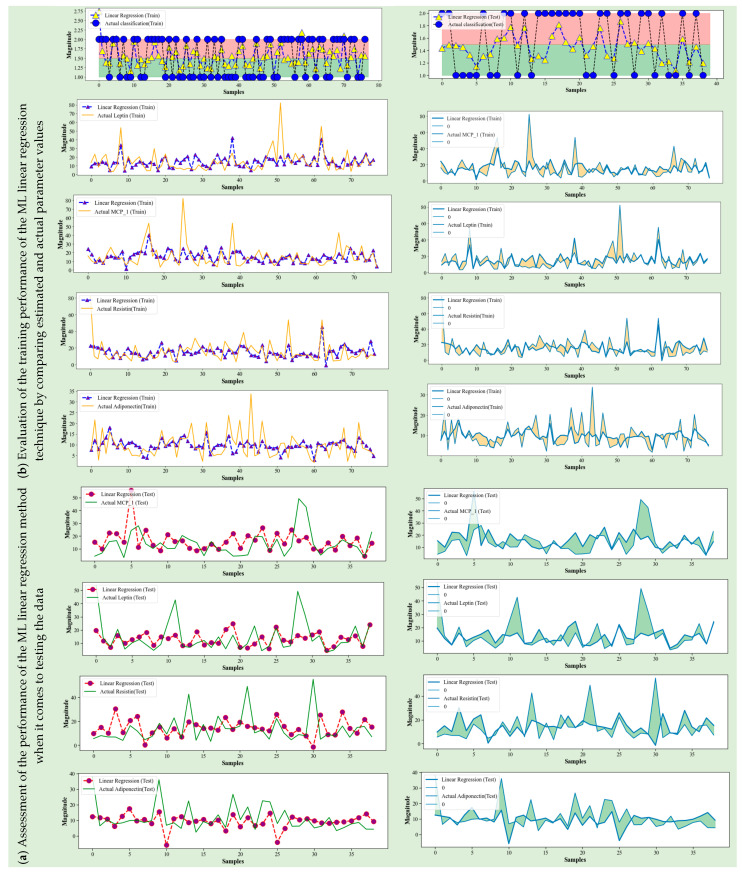
Machine learning-enabled digital twins of cancer based on ml linear regression. (**a**) Evaluation of the training performance of the ML linear regression technique by comparing estimated and actual parameter values. (**b**) Assessment of the performance of the ML linear regression method when it comes to testing the data.

**Figure 4 bioengineering-10-00455-f004:**
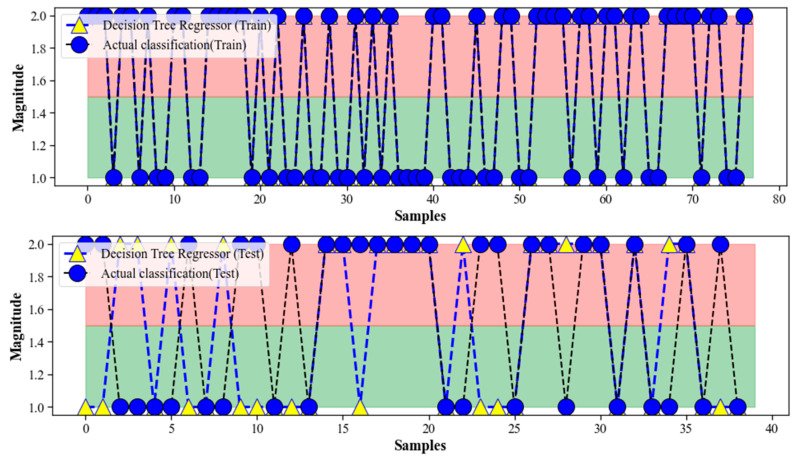
Both the training and testing data are used to demonstrate the process of detecting breast cancer using DTR. The procedure’s efficiency is evaluated using the training data, while its effectiveness is determined by analyzing its performance on the testing data.

**Figure 5 bioengineering-10-00455-f005:**
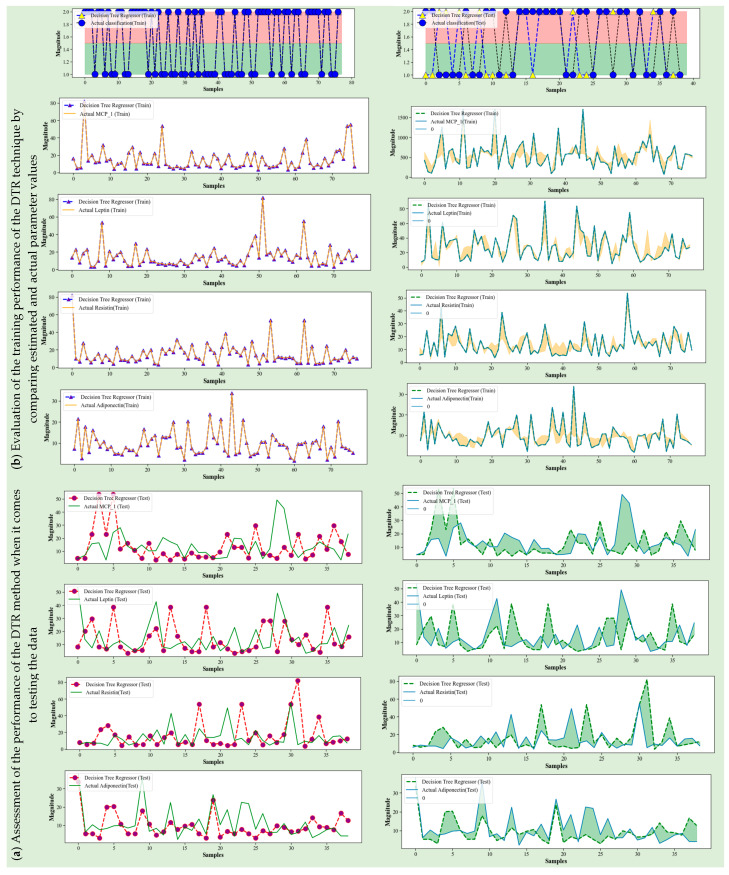
Digital twins of cancer are created through machine learning using ml linear regression. The performance of the DTR technique is evaluated by comparing the estimated and actual parameter values during the training phase. In the testing phase, the effectiveness of the DTR method is assessed by analyzing its performance on the data.

**Figure 6 bioengineering-10-00455-f006:**
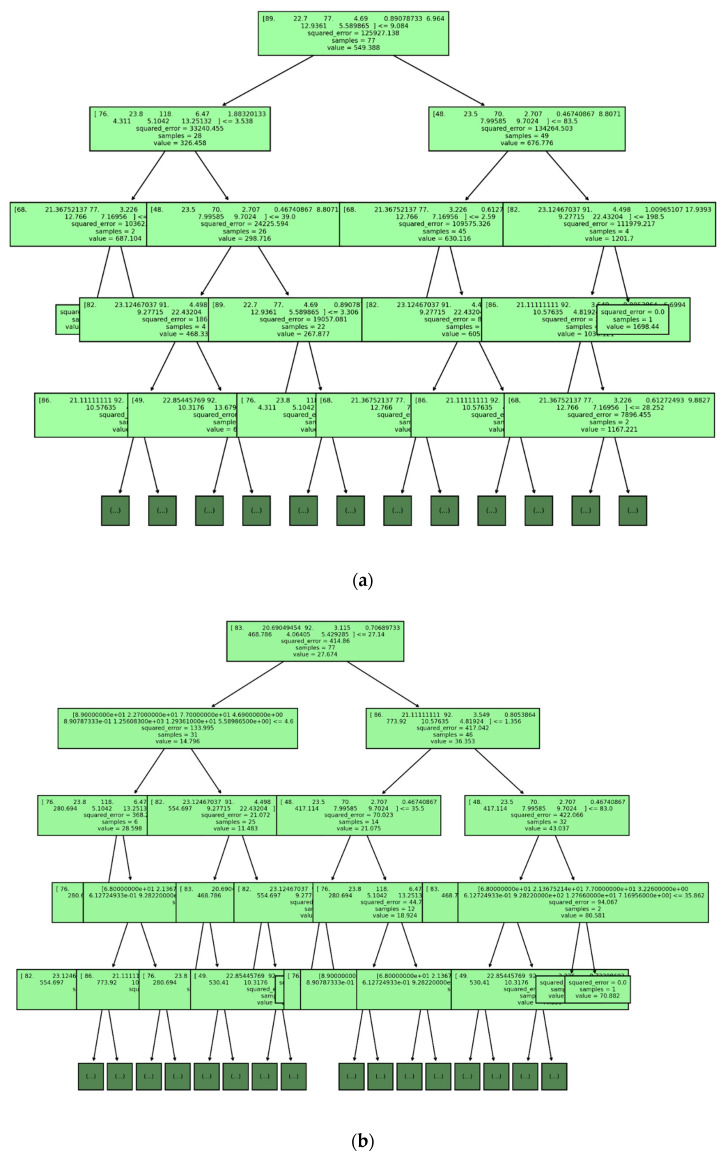
(**a**) The thought process of the prediction model for MCP-1 is illustrated using a decision tree visualization, which displays the feature splits and resulting predictions at each node. (**b**) A decision tree visualization is used to depict the prediction model’s reasoning process for Leptin, indicating the feature splits and resulting predictions at each node.

**Figure 7 bioengineering-10-00455-f007:**
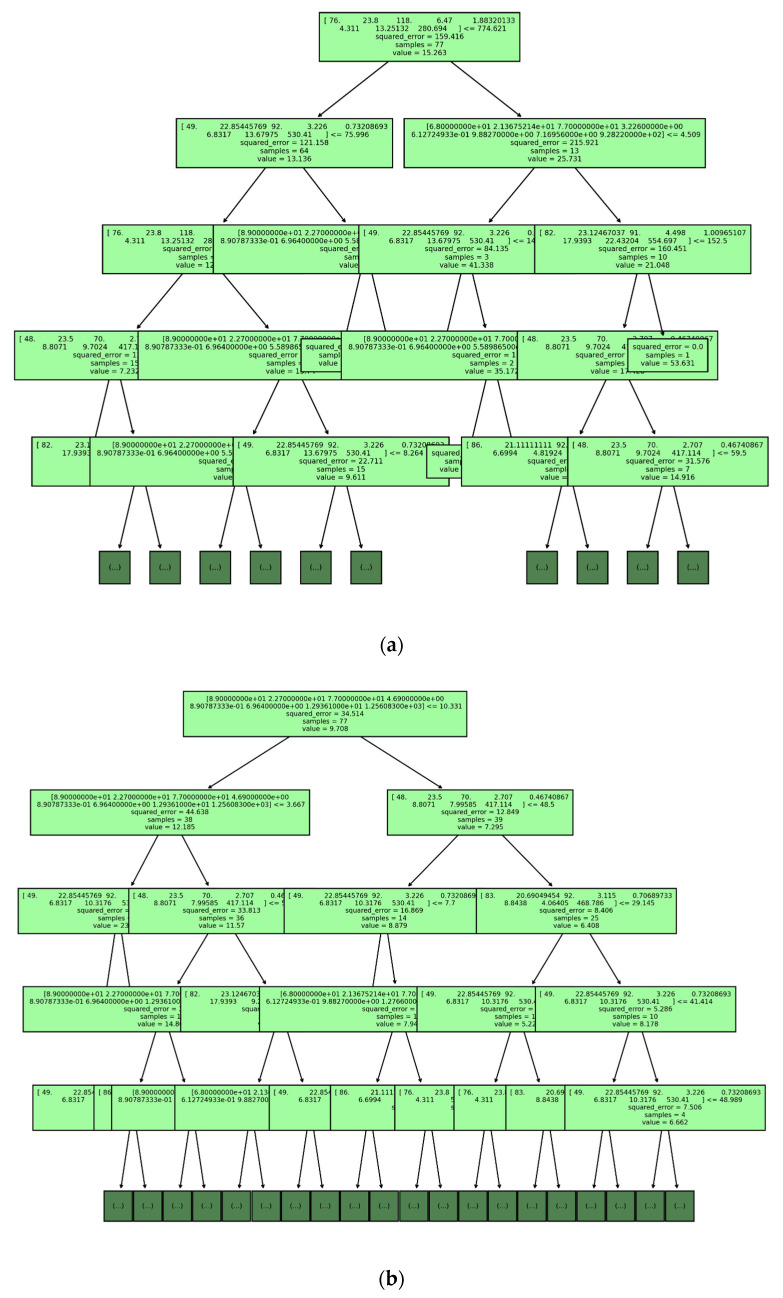
The decision-making process of a prediction model is graphically represented in (**a**,**b**), with a depiction of the various feature divisions and predictions at each stage. The illustration specifically focuses on the features Resistin and Adiponectin.

**Figure 8 bioengineering-10-00455-f008:**
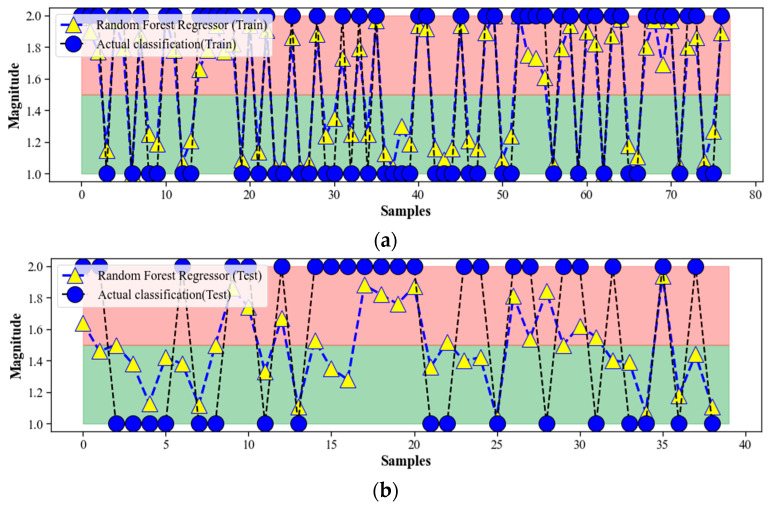
The efficiency of the method in identifying breast cancer by applying it to both training and testing datasets using RFR. The accuracy of the method on the training data is presented in (**a**), while (**b**) demonstrates the evaluation of the technique using the testing data.

**Figure 9 bioengineering-10-00455-f009:**
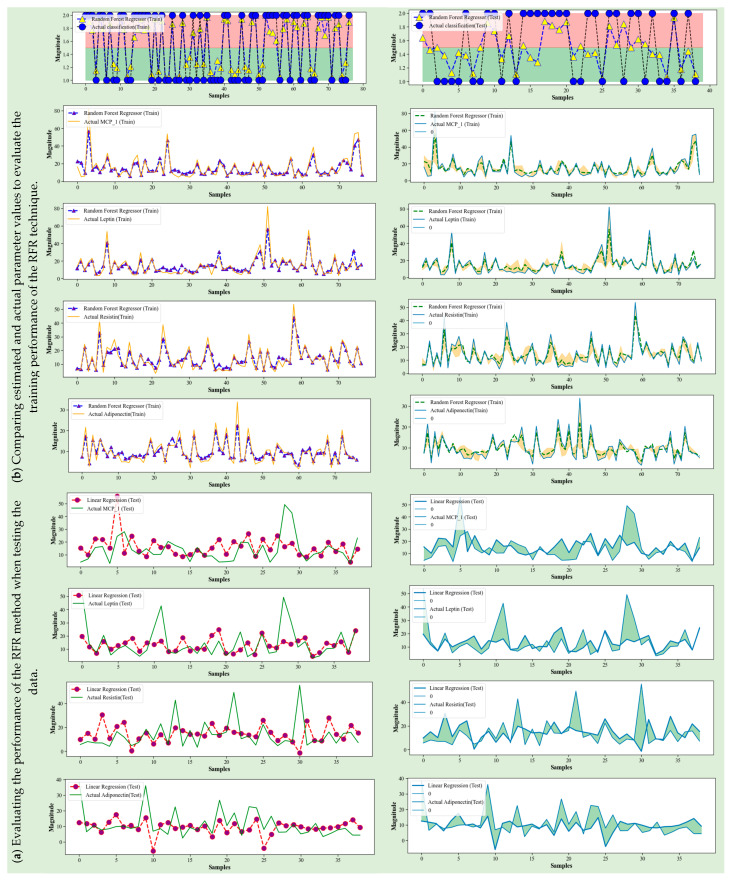
The RFR technique’s ability to detect breast cancer is demonstrated by its application to both the training and testing data. (**a**) The accuracy of the method is displayed for the training data. (**b**) The testing data is utilized to evaluate the technique.

**Figure 10 bioengineering-10-00455-f010:**
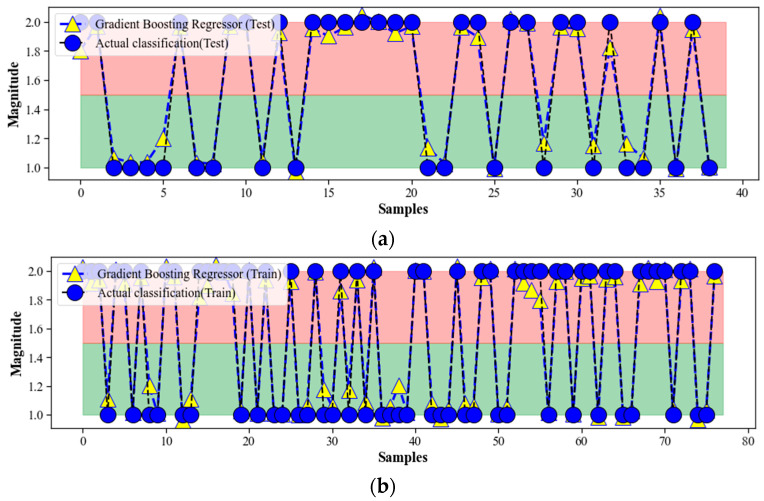
The GBA technique’s ability to detect breast cancer is demonstrated through its application on both training and testing data. (**a**) The accuracy of the technique on the training data is demonstrated. (**b**) The testing data are utilized to evaluate the method.

**Figure 11 bioengineering-10-00455-f011:**
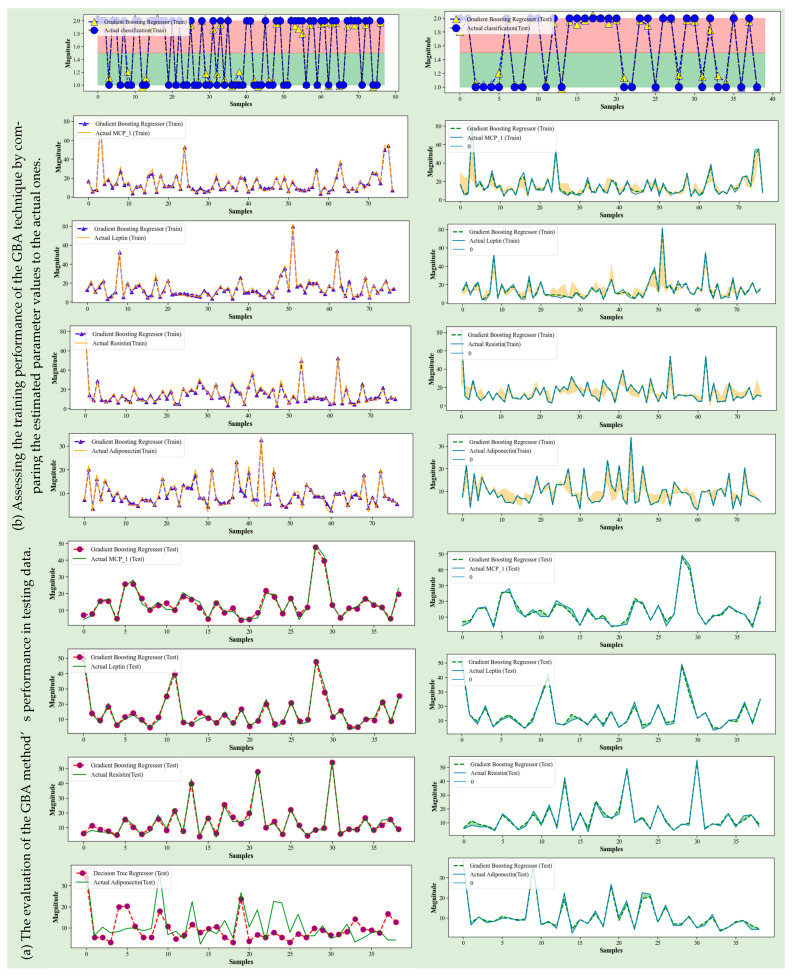
The GBA method’s capacity to identify breast cancer is exhibited by using it on both training and testing data. This means that the accuracy of the method is demonstrated by applying it to the training data, while the testing data are used to assess the approach.

**Figure 12 bioengineering-10-00455-f012:**
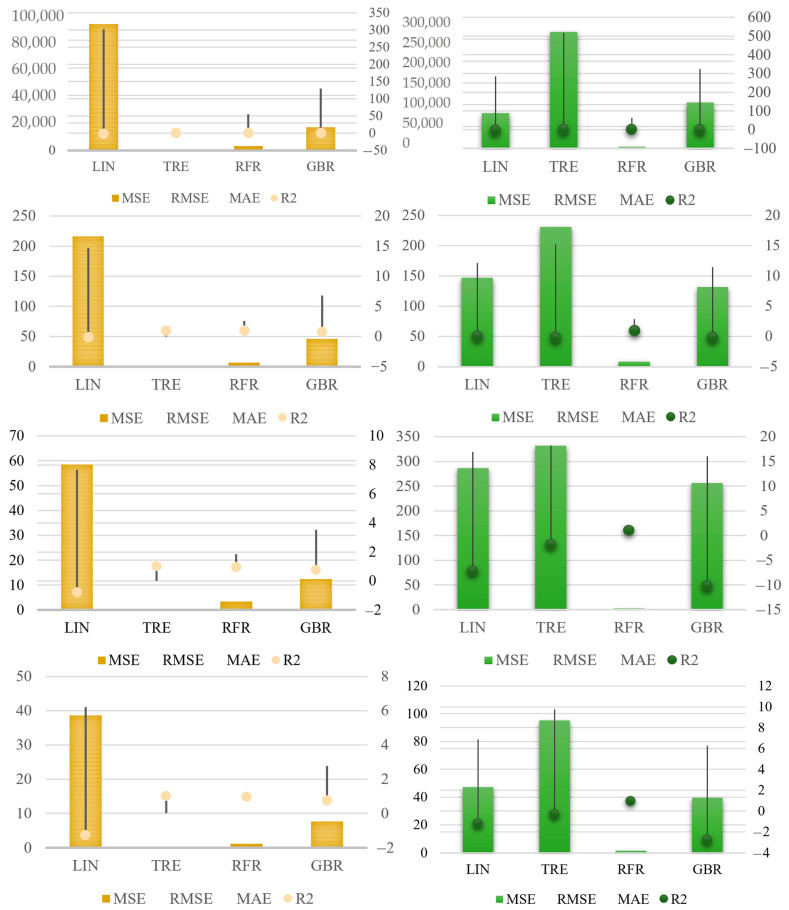
The performance of our prediction model, as evaluated by four common metrics: mean squared error (MSE), root mean squared error (RMSE), mean absolute error (MAE), and coefficient of determination (R2).

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
