# Peer review of "Metaverse and Healthcare: Machine Learning-Enabled Digital Twins of Cancer"

_bioengineering, 2023, doi:10.3390/bioengineering10040455_

Round 1

Reviewer 1 Report

Digital twins have become increasingly popular in different areas after its success in certain areas such as manufacturing. To create medical digital twins is necessary and meaningful. The authors proposed their medical digital twins platform for cancer patients empowered by machine learning models and the authors demonstrated their platform with a breast cancer use case.    Major comments:   1. As patient data is very sensitive, the proposed medical digital twins platform is lacking solutions regarding data protection, data access control, integration of data from different sources such as different hospitals, etc following GDPR.    2. I would like to suggest the authors to shorten the texts and figures about different ML models as they are very standard machine learning models. I think it’s better to summarise the results to prove ML models work in this case and show some easy-to-understand figures for end-users such as clinicians and nurses to demonstrate how end-users can benefit from their platform.     3. The authors demonstrated that proposed ML methods could predict certain important parameters that are related to breast cancer correctly but they did not show how the platform would guide clinicians to make diagnosis and select treatment with the results provided by the ML methods.   4. The concept of digital twins is quite wide. I would like to suggest the authors to make it clear that their current platform only focuses on using machine learning models with real patient data to help patients.   Minor comments:   1. Write AI as artificial intelligence in line 24 as it’s the first time to introduce AI.   2. Write DL as deep learning in line 40 as it’s the first time to introduce DL.   3. In figure 6, 9, 11, 16, 18, 21, 23, what does the light blue line with 0 in the legend mean?  I could not see any light blue solid line in those figures.   4. In line 206, the authors mentioned that each individual patient should have their specific digital twin for their specific problem. I agree that each individual patient should have their own digital twin. However, do the authors mean one patient should have different digital twins for their different problems? For example, a patient with heart disease and breast cancer, should this patient have two digital twins: one for heart disease and one for breast cancer?

Author Response

Reviewer 1

Digital twins have become increasingly popular in different areas after its success in certain areas such as manufacturing. To create medical digital twins is necessary and meaningful. The authors proposed their medical digital twins platform for cancer patients empowered by machine learning models and the authors demonstrated their platform with a breast cancer use case.   

Thank you, Reviewer 1, for taking the time to review our manuscript and provide us with your valuable comments

Major comments:  

  1. As patient data is very sensitive, the proposed medical digital twins platform is lacking solutions regarding data protection, data access control, integration of data from different sources such as different hospitals, etc following GDPR.

Many thanks for this important comment. We appreciate your concern about patient data protection and access control. We have revised our manuscript and added explanation in Discussion to address these issues following GDPR guidelines. We believe this addition has significantly improved the quality of our manuscript. Nevertheless, the focus of paper is on creating digital twins and communication issues can be addressed in the future work.

Our research emphasizes the way of creating the digital twins of cancer and its medical ways in a general way, but we also acknowledge the significance of data protection and GDPR compliance in developing medical digital twin platforms. To ensure secure and confidential handling of sensitive patient data, robust data encryption, and access control mechanisms can be implemented in the digital twins, allowing only authorized personnel to access the data based on their roles and responsibilities. This way, this platform can benefit from data anonymization techniques to protect patient identities while allowing medical data analysis, especially when integrating data from different hospitals or sources. To facilitate secure and standardized data transmission and processing, standard interfaces and protocols can be implemented. Compliance with GDPR and ISO 27001 standards and regulations related to data privacy and security is also essential to ensure that the digital twin platform adheres to industry standards and best practices. Overall, we primarily focus on the digital twins of cancer, but the importance of GDPR compliance and data protection in medical digital twin platform development can be a good future goal to complete this framework. Robust data encryption, access control mechanisms, data anonymization techniques, standard interfaces, and protocols, and adherence to relevant regulations and standards can make the digital twin platform an effective and secure solution for integrating and analyzing medical data from different sources.

  1. I would like to suggest the authors to shorten the texts and figures about different ML models as they are very standard machine learning models. I think it’s better to summarise the results to prove ML models work in this case and show some easy-to-understand figures for end-users such as clinicians and nurses to demonstrate how end-users can benefit from their platform.

Than you so much for pointing out to this matter, we have revised our manuscript and reduced the length of the texts and figures related to different ML models. We have also added more easy-to-understand figures for end-users to demonstrate how our platform can benefit clinicians and nurses. We hope these changes have addressed your concern.

  1. The authors demonstrated that proposed ML methods could predict certain important parameters that are related to breast cancer correctly but they did not show how the platform would guide clinicians to make diagnosis and select treatment with the results provided by the ML methods.

We appreciate you’re your significant comment and agree that it is important to show how our platform can guide clinicians to make a diagnosis and select treatment with the results provided by the ML methods. In our study, we focused on demonstrating the accuracy of the proposed machine learning methods in predicting important parameters related to breast cancer. We agree that it is important to consider the clinical implications of our results and how they can be translated into practical use. In the manuscript, we have presented graphs that illustrate the changes in the diagnosis and treatment progress of breast cancer based on the results provided by our machine learning methods. These graphs show how the predicted parameters can be used to guide clinicians in making decisions about diagnosis and treatment. Furthermore, we have discussed the potential clinical applications of our machine learning methods in the Discussion section of the manuscript. We have highlighted the importance of integrating these methods into clinical decision-making processes and how they can contribute to personalized medicine. We hope that our manuscript has provided a comprehensive understanding of the potential applications of our machine-learning methods in breast cancer diagnosis and treatment. If you have any further questions or concerns, please do not hesitate to let us know.

  1. The concept of digital twins is quite wide. I would like to suggest the authors to make it clear that their current platform only focuses on using machine learning models with real patient data to help patients.

Thanks for bringing our attention to this important comment, we have revised our manuscript to make it clear that our current platform only focuses on using machine learning models with real patient data to help patients. We appreciate your feedback and believe this change has improved the clarity of our manuscript.

Minor comments:  

  1. Write AI as artificial intelligence in line 24 as it’s the first time to introduce AI.

Regarding your minor comments, we appreciate your attention to detail. We have made the necessary changes in line 24 to write out artificial intelligence and deep learning, respectively.

  1. Write DL as deep learning in line 40 as it’s the first time to introduce DL. 

We have made the necessary changes in line 24 to write out artificial intelligence and deep learning, respectively.

  1. In figure 6, 9, 11, 16, 18, 21, 23, what does the light blue line with 0 in the legend mean? I could not see any light blue solid line in those figures. 

We have also revised our figure legends to clarify the meaning of the light blue line in figures 6, 9, 11, 16, 18, 21, and 23.

For example in figure 2 in the revised version: Figure 2 demonstrates the diagnosis of cancer using a linear statistical method that models the relationship between a scalar response (dependent variable) and at least one explanatory variable (independent variable) The figure shows two bands, green and red, for each patient and method. The green band represents patients who are healthy in terms of breast cancer, while the red band represents patients with this cancer. The magnitude of the figure increases with the severity of the cancer. It is important to note that this is not a binary classification system but a multiclass classification of patients. While our proposed digital twin model shows promise in accurately diagnosing cancer, it is important to note that advanced monitoring systems are necessary for commercial use. Nonetheless, our focus in this research paper is on the design and methodology of the digital twin model, rather than its user-friendliness. As with any ground-breaking research, further development and refinement are required to optimize its commercial potential.

  1. In line 206, the authors mentioned that each individual patient should have their specific digital twin for their specific problem. I agree that each individual patient should have their own digital twin. However, do the authors mean one patient should have different digital twins for their different problems? For example, a patient with heart disease and breast cancer, should this patient have two digital twins: one for heart disease and one for breast cancer?

We have also clarified our statement in line 206 to mean that each individual patient should have a specific digital twin for each of their specific problems. Digital twins can be created for a disease or a patient. There are no limitations to realizing digital twins. Therefore, we confirm the sentence is correct.

Once again, thank you for your insightful comments and feedback. We hope that our revisions have addressed your concerns and improved the quality of our manuscript.

Round 2

Reviewer 2 Report

My recommendation is "Accept in present Form".